# Using youth-engaged research methods to develop a measure of disordered eating in transgender, non-binary, and gender-diverse youth: Research protocol

**An Pham**[1]* , **Zoe Webster**[1] , **Melissa-Irene Jackson**[1‡], **Melanie Bean**[1‡],
**Maria Thomson**[2‡], **Suzanne Mazzeo**[3‡], **Kym Ahrens**[4‡]

1 Department of Pediatrics, Virginia Commonwealth University, Richmond, Virginia, United States of America, 2 Department of Health Behavior and Policy, Virginia Commonwealth University, Richmond, Virginia, United States of America, 3 Department of Psychology, Virginia Commonwealth University, Richmond, Virginia, United States of America, 4 Department of Pediatrics, University of Washington, Seattle, Washington, United States of America

☯ These authors contributed equally to this work.
‡ MIJ, MB, MT, SM and KA also contributed equally to this work.
* utanpham@gmail.com

**Data Availability Statement:** Due to the sensitive nature and political targeting of our youth research

## Abstract

Transgender, non-binary, and gender-diverse (TNG) youth experience disordered eating behaviors (DEBs) for reasons unique to their sociocultural positioning and the specific challenges they face, including gender dysphoria and societal beauty expectations of gender, cissexism, and lack of access to gender affirming medical care. The prevalence of DEBs is considerably and consistently higher in TNG youth compared to their cisgender peers. Nonetheless, there are no DEBs measures tailored to this population. Although the field of TNG DEBs research is quickly growing, gaps in knowledge remain, due, in part, to a lack of TNG input on research protocols focused on this population, and underrepresentation of TNG people in DEBs research. The goal of this research protocol is to develop and evaluate a community informed DEBs measure specific for TNG youth. We will implement youth-engaged research methods to create affirming, inclusive research protocols and optimize recruitment of subpopulations of TNG youth historically excluded from research (i.e., trans-feminine youth of color and non-binary youth). A TNG youth advisory board of 5–7 members will participate in all research activities, including developing recruitment matrices, conducting qualitative analyses, developing survey items, interpreting results, and disseminating the scientific findings.

## Introduction

Transgender, nonbinary, and gender-diverse (TNG) youth face stigma due to the marginalization of their gender identities [1–3]. TNG youth also have increased vulnerability to body dissatisfaction due to pubertal changes and development of secondary sexual characteristics that

population, data will not be made publicly available. When the study is completed and published, de-identified research data will be made available upon request for researchers who meet criteria and have created a data sharing agreement with a scientifically appropriate plan for data usage.

**Funding:** Dr. Pham acknowledges her support by NIH/NIMH K23 MH134111. The project is in part also supported by CTSA award No. UM1TR004360 from the National Center for Advancing Translational Sciences. Its contents are solely the responsibility of the authors and not necessarily represent official views of the National Center for Advancing Translational Sciences or the National Institutes of Health. The funders had/will have no role in study design, data collection and analysis, decision to publish, or preparation of the manuscript.

**Competing interests:** The authors have declared that no competing interests exist.

might be misaligned with their gender identity, which may be exacerbated by a youth's inability to access gender-affirming medical care (i.e., puberty blockers, gender-affirming hormones). Moreover, compared to the general population, TNG youth experience higher rates of subthreshold eating disorder symptomatology, also known as disordered eating behaviors (DEBs) [4, 5]. DEBs include binge eating, fasting, and purging [4–9]. Based upon data from the general population, DEBs are associated with significant physical and emotional impairment (i.e., changes in menstruation and fertility, substance use, depressive symptoms) and often precede the onset of eating disorders [10–15]. Furthermore, 30–51% of TNG youth report suicidal ideation, and TNG young adults with eating disorders attempt suicide at higher rates than their TNG peers without eating disorders [10, 16, 17]. Despite these health disparities among TNG youth, DEBs and eating disorder research has historically focused on white, cisgender young adult women [18, 19].

Currently, there are no DEBs measures specific to TNG youth. The only DEBs measure currently validated with TNG young adults is the Eating Disorder Examination–Questionnaire Short (EDE-QS). However, this measure has been critiqued for failing to adequately consider the unique lived experiences of TNG people [20]. For example, TNG youth have identified gender dysphoria and societal beauty expectations of gender as factors contributing to their DEBs symptomatology [21]. No current measures assess the attitudes and cognitions underlying TNG youth's DEBs. This creates a major barrier to gender-affirming care and accurate diagnosis of EDs and DEBs in trans youth. A tailored instrument that assesses the specific behaviors, attitudes, and cognitions contributing to TNG youths' DEBs is needed to advance research on this understudied topic, address underdiagnosis, and improve clinical care.

Research investigating DEBs among TNG individuals is a quickly growing field of investigation, but knowledge gaps persist due to underrepresentation of TNG participants in research. Inclusive recruitment and engagement practices are critical to the expansion of TNG youth research; however, TNG youth may avoid research participation due to barriers associated with their gender identity. For example, obtaining guardian consent might not be feasible for TNG youth under the age of 18 who are not open about their gender identity and/or do not live in a supportive home environment [22]. Moreover, the increase in anti-transgender legislation over the past 5 years may affect TNG youth research participation [23]. Additionally, gender-affirming youth clinics have seen a shift towards transmasculine patients [24] and TNG youth research samples are commonly skewed towards white, transmasculine youth with parental support and access to gender-affirming care in urban areas [2, 8, 17, 20, 21, 25]. Approximately 38.5% of TNG adults are transfeminine, 35.9% transmasculine, and 25.6% gender non-conforming [26]. Estimates of TNG population are not fully accurate due to the broad range of terms surveys use to describe gender identity, lack of TNG data collection in population-based surveys, and difficulty creating gender identity categories (as individuals can identify with multiple gender identities). Despite this, the discrepancy between the proportion of transfeminine people and their underrepresentation in research is a critical gap in knowledge because transfeminine people, particularly transfeminine people of color, experience unique physical and mental health challenges [27–30]. Race/ethnicity, access to resources, and gender identity have an impact on TNG youth experiences [31, 32]. As a result, if researchers are unable to recruit samples that represent TNG youth diversity, validity can be compromised. A critical step to diversifying TNG youth research participation is to explore barriers and motivators to this behavior, particularly within minoritized subpopulations of TNG youth who have historically been excluded from scientific study.

In addition to a lack of TNG youth recruitment in research, TNG input is largely absent from TNG research protocols. Youth-engaged research methods aim to systematically improve research participation and protocols by directly incorporating input from the population of

interest [33–35]. Youth-engaged research methods have the potential to create research processes and measures that are suited to TNG individuals' experiences and could increase recruitment and engagement among racially and ethnically diverse TNG youth across the age and gender identity spectrum. The objective of the current research protocol is to use youth-engaged research methods to develop a DEBs measure that reflects the unique needs of all TNG youth, including under-researched subpopulations such as racial and ethnic minorities, and transfeminine and non-binary individuals.

# Materials and methods

## Study aims, design, and setting

This research project has four phases:

### Phase 1: Create a TNG youth advisory board (YAB).

This research protocol will use a community-engaged approach to create a TNG youth-centered measure of disordered eating behaviors, attitudes, and cognitions. To ensure the entire research process is tailored to the strengths and needs of TNG youth, we will create a board of 5–7 TNG youth advisors with a history of DEBs. Details on responsibilities can be found throughout the remainder of the protocol. In general, participation will include 1–2 virtual YAB meetings a month, review and feedback on research documents (i.e., recruitment flyers, interview scripts), qualitative analyses, and generation of measure items. We estimate 3–4 hours per month for each YAB member. For every 3 months of participation, YAB members will be given a $125 honorarium. Additionally, YAB members will be paid $20 for each transcript they code during qualitative analysis in Phases 2 and 3. This compensation plan was based off feedback from community members and other researchers conducting YABs with TNG youth. All YAB members will also be given the opportunity to co-author any manuscripts written from the research activities of this protocol. These responsibilities and expectations will be given to YAB members prior to written consent. We will implement a co-design approach to YAB meetings and participation. We have collected strategies from literature, online community advisory board toolkits, and our experience from successful community advisory boards on other research projects [36–38]. Strategies include co-creation of a YAB community agreement, creating distinct activities with a definitive process for providing feedback, asynchronous participation strategies to allow YAB members to complete tasks in between meetings, and monthly feedback and partnership building evaluation (see Data Collection for further details). The YAB will be facilitated by a research assistant, a research team member who identifies as TNG, who will be trained prior to the initiation of any YAB activities. This study protocol provides a foundation for this multi-phase study and will be modified as YAB members provide feedback throughout Phases 2–4.

### Phase 2: Apply qualitative methods to identify facilitators and barriers to TNG youths' research participation.

We will conduct semi-structured interviews to explore the perspectives of TNG youths regarding participation in community-engaged, survey, and qualitative research. We will over-sample underrepresented TNG identities (i.e., TNG youth of color, transfeminine and non-binary youth). We have chosen a sample size of 20 participants based on the focus of the research question, methods of data collection (see Data Collection), sample diversity, and depth of data likely generated from each participant [39]. Given the lack of TNG youth diversity across most research topics, the goal of this research activity is to understand barriers and facilitators to participation in all types of research. Because the focus of Phases 3 and 4 is on

DEBs, the interview script will also include questions specific to participation in research on DEBs. Findings will inform recruitment protocols applied in Phases 3 and 4.

Phase 3: Generate and evaluate initial items of DEBs measure for TNG youth.

We will conduct 3 focus groups addressing DEBs in TNG youth (~6–8 TNG youth per group). The final sample size and sociodemographic makeup of each focus group will be determined by the YAB. YAB members will individually generate a list of DEBs items based on the focus group data. The YAB will then discuss and rate all items and narrow the list into a preliminary DEBs measure of behaviors, attitudes, and cognitions. We will then conduct cognitive interviews to refine the DEBs measure and ensure all items are clear, understandable, and affirming to TNG youth. We will implement a hybrid model of a think-aloud approach and verbal probing [40]. Cognitive interviews will be conducted in phases of 2–3 interviews with subsequent modifications to the DEBs measure based on patterns of cognitive processing, consistencies, and inconsistencies of the interview results within that phase. We anticipate conducting a total of approximately 10 cognitive interviews.

Phase 4: Examine psychometric properties of the DEBs measure and investigate relations between DEBs and TNG risk and protective factors.

Once refined, we will test (via exploratory factor analysis, EFA) and confirm (via confirmatory factor analysis, CFA) the structure, reliability, validity, and invariance of the DEBs measure. We will apply data from Phase 2 and recruit a diverse sample of TNG youth (n≈500) to evaluate scale items and test (n≈250) and confirm (n≈250) the structure of the scale [41]. In addition to confirming scale structure, measurement invariance in relation to sex assigned at birth, gender identity, and race/ethnicity will be evaluated. Based on extant literature, we will also explore associations between DEBs and TNG-specific risk and protective factors (i.e., mental health diagnoses, socioeconomic status, gender minority stress, peer/family support, access to affirming medical care). We hypothesize that certain behaviors, attitudes, and cognitions will: 1) be more strongly correlated with higher levels of depression and anxiety, and 2) differ depending on an individual's exposure to stigma (i.e., gender-related discrimination) and access to external support (i.e., peers/family members support, gender-affirming medical care). The final DEBs measure for TNG youth will be made freely available for download on a research website.

The first phase of this protocol was approved by Virginia Commonwealth University's IRB (HM20026881). As all subsequent phases are dependent on Phase 1 findings, we will submit amendments and no further activities beyond Phase 1 will be conducted prior to IRB approval.

## Recruitment

We will apply findings from Phase 2 (qualitative interviews to identify facilitators and barriers to research participation) to recruitment protocols for Phases 3 (develop DEBs measure) and 4 (psychometric analyses of DEBs measure). For Phases 2–4, our YAB and a community consultant with expertise in DEBs in TNG youth will help create a recruitment stratification matrix (age x gender identity x race/ethnicity). Additionally, for Phases 2–4, as recommended for sexual and gender minority youth, we will seek a waiver of parental consent from Virginia Commonwealth University's IRB to decrease barriers and facilitate participation of TNG youth with varying degrees of guardian support [22, 42]. Informed consent for all Phases will be written. Because Phase 1 will require parental consent, we will obtain written informed consent from both the participant and legal guardian for any YAB candidate under 18 years old.

Research on TNG youth has primarily sampled transmasculine youth with access to gender-affirming medical care [17, 24]. As such, recruitment for all Phases will occur in both

clinical and non-clinical settings to create a diverse sample of participants regarding gender identity, resources, level of support around gender identity, and access to gender-affirming medical care. Clinical recruitment settings will include gender-affirming medical clinics and local and national listservs of gender-affirming medical and mental health providers. Non-clinical recruitment settings will include local and national TNG serving organizations. To avoid some of the limitations seen in past efforts researching the needs of TNG individuals, in addition to prioritizing input from the YAB, the process for recruitment and each subsequent research activity will be conducted with input and guidance from members of the TNG community who also have experience and knowledge in research and/or DEBs. Specifically for recruitment, we will seek guidance from the YAB, community consultant, and TNG community members to identify other appropriate organizations and opportunities for recruitment, particularly for underrepresented TNG subpopulations.

## Inclusion and exclusion criteria

Table 1 includes inclusion criteria for all research phases. All phases require English proficiency and gender identity different than assigned sex at birth. Inclusion criteria for Phases 1 and 3a include a history of DEBs to facilitate discussion specifically on DEBs. A history of DEBs will be satisfied if a potential participant selects "yes" to questions 1, 2, 7, 8, 9, or 10 on the EDE-QS (measure of DEBs that has been validated in a sample of 71 TNG young adults) [20]. We have chosen these specific questions from the EDE-QS because they measure a disordered eating behavior (i.e., food restriction, laxative use, binge eating, compulsive exercising) and not a cognition (i.e., weight or shape has influenced how one thinks about themselves as a person). Because cognitions related to body/shape dissatisfaction might be secondary to gender dysphoria and societal beauty expectations of gender and not disordered eating, we only included questions measuring behaviors.

Phase 2 aim has broader application than TNG youth with DEBs. The interview topic is barriers and facilitators to research participation for all TNG youth and is intended to be generalizable to non-DEB samples. Therefore, Phase 2 does not include a history of DEBs as an inclusion criterion. The purpose of Phases 3b and 4 are to develop and examine the psychometric properties of a measure to screen for DEBs in all TNG youth, so inclusion criteria do not specifically include a history of DEBs.

We have chosen an age range of 12–21 for the development of a DEBs measure in Phases 3–4 because eating disorders peak during adolescence and young adulthood [43, 44].

**Table 1. Unique inclusion criteria for each research phase.**

| Research activity | Unique inclusion criteria (all phases require English proficiency and gender identity different than assigned sex at birth) |
| --- | --- |
| Phase 1: TNG YAB | • Age 16–21.<br>• History of DEBs |
| Phase 2: Individual interviews on facilitators and barriers to research participation | • Age 12–21 |
| Phase 3a: Focus groups on DEBs among TNG youth to develop a DEBs measure | • Age 12–21<br>• History of DEBs |
| Phase 3b: Cognitive interviews to test DEBs measure items | • Age 12–21 |
| Phase 4: Surveys for DEBs measure evaluation | • Age 12–21 |

## Data collection

Table 2 contains an overview for each phase. Over the entire research protocol, YAB members will meet with the research assistant 1–2 times per month and participate in all aspects of Phases 2–4, from developing recruitment matrices, to conducting qualitative analyses, to interpreting and disseminating results. To evaluate participation and engagement of YAB members, we will measure number of meetings attended, involvement in research processes (i.e., number of research materials reviewed, number of qualitative scripts coded, and generation of measure items), and partnership building with the brief version of the Research Engagement Survey Tool (REST) for each YAB member monthly [45]. This monthly data collection will also include opportunity for YAB members to individually provide positive and negative feedback on the research team and YAB research processes. To optimize recruitment and retention, we will review these data, meet with youth advisory board members with lower engagement to discuss bidirectional feedback, and modify research protocols accordingly.

For qualitative methods in Phases 2 and 3a, the research assistant and community consultant will conduct semi-structured interviews by Zoom video conference. Participant chosen/preferred first name and first letter of last name and contact information will be entered in a HIPAA secure REDCap database and attached to the participant's unique study ID number [46, 47]. This database will be separate from all other study information databases with data that is only identified by the study ID number. The interviewer will confirm that the TNG youth participant is in a private and comfortable location before initiating any study-related conversations. We will obtain permission and preference of contact by phone or email when scheduling the interview. All interviews will be audio-recorded and transcribed. The audio recordings/transcripts will only contain the participant's study ID number and no identifying information. Each audio recording will be deleted immediately after study staff checks the accuracy of transcription.

To develop an initial DEBs measure in Phase 3, YAB members will use the qualitative data from Phase 2 and individually generate a list of items. YAB members will then come together, discuss and rate all generated items, and then narrow the list of items into a preliminary DEBs

**Table 2. Research process overview for each phase.**

| Research Activity | Data Collection | Compensation | Analyses |
|---|---|---|---|
| Phase 1: TNG YAB | • Number of meetings attended<br>• Research materials reviewed<br>• Community Engagement Measure [48] | • $125 for every 3 months of participation<br>• $20 for each transcript coded in Phases 2 & 3 | • Descriptive analyses of participation and engagement<br>• Summary of findings will contribute to modification of research protocols to improve research processes and YAB retention |
| Phase 2: Individual interviews on facilitators and barriers to research participation | • Qualitative interview transcripts | • $30 | • Braun and Clarke thematic approach [49] |
| Phase 3a: Focus groups on DEBs among TNG youth to develop a DEBs measure | • Qualitative interview transcripts | • $30 | • Braun and Clarke thematic approach [49] |
| Phase 3b: Cognitive interviews to test DEBs measure items | • Cognitive interview transcript | • $30 | • Create item summaries and determine patterns |
| Phase 4: Surveys for DEBs measure evaluation | • One-time online survey | • $20 | • Psychometric analyses:<br>○ Exploratory factor analysis<br>○ Confirmatory factor analysis<br>○ Internal consistency<br>○ Validity<br>○ Invariance analysis<br>○ Regression analysis |

measure. The research assistant will conduct cognitive interviews in Phase 3b. They will implement a hybrid model of a think-aloud approach and verbal probing [40]. After each item's think-aloud portion, the interviewer will ask open-ended probes. Cognitive interviews will be audio-recorded and transcribed.

In Phase 4, we will collect the following data in one-time, online REDCap [46, 47] surveys:

1. DEBs measure developed in Phase 3

2. Eating Disorder Examination Questionnaire-Short (EDE-QS): 12-item measure of disordered eating behaviors in the past week validated in a sample of 71 TNG individuals [20].

3. Patient Health Questionnaire-9 (PHQ-9): measure of depressive symptoms in the past two weeks validated in adolescents [50].

4. General Anxiety Disorder-7 (GAD-7): measure of anxiety symptoms in the past two weeks validated in adolescents [51].

5. Gender Minority Stress and Resiliency (GMSR): measure of gender-related discrimination, gender-related rejection, gender-related victimization, and non-affirmation of gender identity [52].

6. Multi-Dimensional Scale of Perceived Social Support (MSPSS): measure of perceptions of support across three levels: family, friends, and significant others [53].

7. Brief Resilience Scale (BRS): measure of an individual's ability to bounce back after stressful situations [54].

8. Sociodemographic survey: questions assessing an individual's age, sex assigned at birth, intersex [55], gender identity, race/ethnicity, socioeconomic status (education, income, food insecurity), medical affirmation (gender-affirming medications and surgeries) and desire for medical affirmation, congruence of gender identity and expression, history of mental health and eating disorder diagnoses.

This list of measures will be modified and finalized with feedback from the community consultant, YAB, and community stakeholders.

## Analyses

For Phases 2 and 3a, we will analyze qualitative data using Braun and Clarke's reflexive thematic analysis approach [49]. Steps of an iterative thematic analysis include: 1. Familiarizing oneself with the data, 2. Generating initial codes, 3. Searching for themes, 4. Reviewing themes, 5. Defining and naming themes, and 6. Writing the report [49]. The PI and research assistant will train YAB members to participate in reflexive thematic analysis. All TNG youth members, each paired with trained research staff, will code transcripts. Coders will independently read two transcripts to establish familiarity (Step 1). Coders will then use the first five transcripts to develop a codebook by independently coding the same transcripts, comparing results, and adjusting code definitions as needed (Step 2). Once a codebook is developed, each transcript will be coded by at least 2 YAB members (with the assistance of their research team partner). The research team will compare codes for each transcript and all discrepancies will be resolved by the PI and research assistant. Coders will meet with the PI and research assistant weekly to bi-weekly by Zoom video conference to discuss any questions on codes, code definitions, and coding process. Using a data driven, inductive approach, the research team will work with the YAB members to review patterns and establish themes (Step 3). The PI and research assistant

will meet with YAB members weekly to bi-weekly to identify and confirm persistent themes (Steps 4–5).

To determine whether each item of the initial DEBs measure developed in Phase 3a is: 1) clear and understandable; 2) affirming and inclusive of TNG youth, the research team will review cognitive interview transcript data. We will create item summaries and determine patterns of cognitive processing, consistencies, and inconsistencies.

In Phase 4, we will test and confirm structure, reliability, validity, and invariance of the DEBs measure. To test and confirm structure, we will randomly split the DEBs measure data into two community samples of 250 and: 1) explore the measure's factor structure and further refine the measure with EFA with Sample 1 and 2) conduct maximum-likelihood based CFA with Sample 2 to cross-validate the structure identified with Sample 1 and create a final version of the measure [56, 57]. The reliability of the DEBs measure will be evaluated with internal consistency. To examine validity, we will compare DEBs scores among participants with and without DEBs based on two different methods of operationalizing DEBs. The first method is participant self-report of ever being diagnosed with an eating disorder and the second method is the EDE-QS score [20]. We will conduct invariance analyses of the best fitting models (form, loadings, and intercepts) to test whether the DEBs measure is invariant across gender identities or sex assigned at birth (i.e., capacity to menstruate) [58]. Using our newly validated measure, we will examine relations between DEBs and well-established mental health disparities and risk and protective factors. We will conduct stepwise regression analyses with DEBs and mental health outcomes (depression, anxiety) and risk and protective factors (gender minority stress, peer/family support, access to affirming medical care, resilience).

## Discussion

This multi-phase study protocol applies youth-engaged research methods and uses inclusive, affirming research processes to explore both barriers and motivators to health research in TNG youth. These data will subsequently be applied to recruit under researched subpopulations of TNG youth (i.e., transfeminine youth, TNG youth of color, and TNG youth without family support and/or access to gender-affirming medical care) in order to create a tailored, evidence-based DEBs measure for TNG youth. The overall objective is to advance this scientific area and improve clinical care for a marginalized gender minority population. Findings have the potential to transform clinical practice for a stigmatized population by developing a DEBs measure that will inform prevention and intervention efforts for TNG youth.

The use of YAB is becoming more common in TNG research and we are committed to incorporating TNG youth input in our research on DEBs [42]. The external validity of this study is enhanced by its incorporation of TNG youth input into the experimental designs [59, 60]. We will construct a DEBs measure using a high degree of TNG youth input in all protocol aspects. To do so, we will partner with a YAB and community consultant with expertise in TNG DEBs and employ youth-engaged research methods from research design to dissemination. Efforts were also made to recruit research staff, specifically, the research assistant and community consultant, with lived experience as being TNG [42]. External validity will also be strengthened with national recruitment, stratified recruitment matrices, and oversampling of historically excluded subpopulations of TNG youth to ensure a racially/ethnically diverse sample across the gender identity spectrum.

Although results from Phase 2 will provide insight into optimal recruitment and retention processes for historically excluded subpopulations of TNG youth, stigma and psychosocial barriers will continue to exist [61–64]. Particularly with recent changes in legislation affecting TNG youth (i.e., gender-affirming medical bans for youth, inability to use a restroom that

aligns with a youth's gender identity), youth may not feel safe participating in TNG related research [23]. Moreover, because participants will be engaging in discussions or completing a survey in English, English proficiency is an inclusion criterion for all phases of our protocol. As a result, our study samples may not fully reflect the diversity of TNG youth; however, by: 1) including a research activity that explores barriers and facilitators to research participation, 2) oversampling underrepresented subpopulations of TNG youth in all research phases, and 3) requesting a waiver of parental consent in Phases 2–4, we have made sample diversity a priority throughout our entire multi-phase research protocol. Additionally, we are recruiting nationally for the YAB and meetings will be held virtually. We anticipate there will be challenges to YAB meetings, specifically, coordinating times that all youth can meet and building working relationships virtually. To address this potential limitation, the research assistant will: 1) conduct YAB meetings outside of business hours, including weekends, 2) allow 2 "drop in" times for one meeting topic so that people in different time zones and/or with different commitments (i.e., school or work) can participate, 3) create team building activities and ice breakers for each YAB meeting, and 4) provide a compensation of $125 for every 3 months of their time and participation and $20 for each code they transcript in Phases 2 and 3.

## Conclusion

TNG youth are at high risk of DEBs and the need for tailored measures of these symptoms is urgent for this population. Yet, DEBs research has largely excluded TNG populations and focused on cisgender women [18, 19]. This study will advance the growing field of research on DEBs in TNG youth. Phase 2 of our multi-phase protocol will diversify health research recruitment of TNG youth and Phases 3 and 4 will generate a DEBs measure specifically for TNG youth that can be used in research and clinical practice. By advancing this field of research and improving clinical care, this study is contributing to an overall goal of promoting equity and reducing mental and physical health disparities for TNG youth. Lastly, this study aims to diversify the research workforce. The PI and research assistant will be training the YAB members to conduct qualitative analysis and the YAB members will also directly be involved in measure item development. We hope that this capacity building with YAB members will promote representation of TNG investigators in research.

## Acknowledgments

Dr. Pham acknowledges her support from the resources provided by CTSA award No. UM1TR004360 from the National Center for Advancing Translational Sciences. Its contents are solely the responsibility of the authors and not necessarily represent official views of the National Center for Advancing Translational Sciences.

## Author Contributions

**Conceptualization:** An Pham, Zoe Webster, Melissa-Irene Jackson, Melanie Bean, Maria Thomson, Suzanne Mazzeo, Kym Ahrens.

**Methodology:** An Pham, Zoe Webster, Melissa-Irene Jackson, Melanie Bean, Maria Thomson, Suzanne Mazzeo, Kym Ahrens.

**Supervision:** Melanie Bean, Maria Thomson, Suzanne Mazzeo, Kym Ahrens.

**Writing – original draft:** An Pham, Zoe Webster.

**Writing – review & editing:** Melissa-Irene Jackson, Melanie Bean, Maria Thomson, Suzanne Mazzeo, Kym Ahrens.

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
