## [Decision Letter · Decision Letter 0]

2 Aug 2024

PONE-D-24-25313Using youth-engaged research methods to develop a measure of disordered eating in transgender, non-binary, and gender-diverse youth: research protocolPLOS ONE

Dear Dr. Pham,

Thank you for submitting your research protocol to PLOS ONE. After careful consideration, we feel that it has merit but does not fully meet PLOS ONE’s publication criteria as it currently stands. Therefore, we invite you to submit a revised version of the protocol that addresses the points raised during the review process.

Please submit your revised protocol by Sep 14 2024 11:59PM. If you will need more time than this to complete your revisions, please reply to this message or contact the journal office at plosone@plos.org. When you're ready to submit your revision, log on to https://www.editorialmanager.com/pone/ and select the 'Submissions Needing Revision' folder to locate your protocol file.

Please include the following items when submitting your revised protocol:A rebuttal letter that responds to each point raised by the academic editor and reviewer(s). You should upload this letter as a separate file labeled 'Response to Reviewers'.A marked-up copy of your manuscript that highlights changes made to the original version. You should upload this as a separate file labeled 'Revised Protocol with Track Changes'.An unmarked version of your revised paper without tracked changes. You should upload this as a separate file labeled 'Protocol'.

We look forward to receiving your revised protocol.

Kind regards,

Christina M. Roberts, M.D., M.P.H.

Academic Editor

PLOS ONE

2. Please provide additional details regarding participant consent. In the ethics statement in the Methods and online submission information, please ensure that you have specified (1) whether consent will be informed and (2) what type you obtain (for instance, written or verbal, and if verbal, how it will be documented and witnessed). If your study includes minors, state whether you obtain consent from parents or guardians. If the need for consent was waived by the ethics committee, please include this information. 

 [Dr. Pham acknowledges her support by NIH/NIMH K23 MH134111. The project is in part also supported by CTSA award No. UM1TR004360 from the National Center for Advancing Translational Sciences. Its contents are solely the responsibility of the authors and not necessarily represent official views of the National Center for Advancing Translational Sciences or the National Institutes of Health. ].  

[Dr. Pham acknowledges her support by NIH/NIMH K23 MH134111. The project is in part also supported by CTSA award No. UM1TR004360 from the National Center for Advancing Translational Sciences. Its contents are solely the responsibility of the authors and not necessarily represent official views of the National Center for Advancing Translational Sciences or the National Institutes of Health.]

 [Dr. Pham acknowledges her support by NIH/NIMH K23 MH134111. The project is in part also supported by CTSA award No. UM1TR004360 from the National Center for Advancing Translational Sciences. Its contents are solely the responsibility of the authors and not necessarily represent official views of the National Center for Advancing Translational Sciences or the National Institutes of Health. ]

7. Please amend your list of authors on the manuscript to ensure that each author is linked to an affiliation. Authors’ affiliations should reflect the institution where the work was done (if authors moved subsequently, you can also list the new affiliation stating “current affiliation:….” as necessary).

8. Please include your full ethics statement in the ‘Methods’ section of your manuscript file. In your statement, please include the full name of the IRB or ethics committee who approved or waived your study, as well as whether or not you obtained informed written or verbal consent. If consent was waived for your study, please include this information in your statement as well. 

9. Please include your tables as part of your main manuscript and remove the individual files. Please note that supplementary tables (should remain/ be uploaded) as separate ""supporting information"" files".

Additional Editor Comments:

Thank you for submitting your research protocol for our review. The reviewers and I are excited about your proposed research. However, the reviewers have several suggestions and requests for clarification. I recommend considering these suggestions and revising your proposed protocol to address these concerns.

Amongst other concerns, the reviewers suggested expanding your literature review and using this additional information to inform overall use of inclusive language, justification of your study, and your description of the qualitative methods you will be using. I have listed links to some of the resources suggested by our reviewers to facilitate access to these materials.

Self-Consent for HIV Prevention Research Involving Sexual and Gender Minority Youth: Reducing Barriers Through Evidence-Based Ethics https://journals.sagepub.com/doi/abs/10.1177/1556264616633963Lessons from a community-based participatory research study with transgender and gender nonconforming youth and their families https://journals.sagepub.com/doi/abs/10.1177/1476750318818875Researching and Working for Transgender Youth: Contexts, Problems and Solutions https://www.mdpi.com/2076-0760/5/3/43A scoping review of good methodological practices in research involving transgender, non-binary, and two-spirit youth. https://www.tandfonline.com/doi/abs/10.1080/19361653.2022.2092576The Inclusion of the LGBTQIA+ Community in Research: a Rapid Scoping Review on Barriers and Facilitators.  https://essay.utwente.nl/97054/1/Strieckmann_MA_BMS.pdfPerceptions of Barriers to and Facilitators of Participation in Health Research Among Transgender People.  https://www.liebertpub.com/doi/full/10.1089/trgh.2016.0023A systematic review of sociodemographic reporting and representation in eating disorder psychotherapy treatment trials in the United States.  https://doi.org/10.1002/eat.23699Differences in all-cause mortality among transgender and non-transgender people enrolled in private insurance.  https://doi.org/10.1215/00703370-9942002To saturate or not to saturate? Questioning data saturation as a useful concept for thematic analysis and sample-size rationales.  https://doi.org/10.1080/2159676X.2019.1704846Virginia Braun & Victoria Clarke (2021). Thematic analysis: A practical guide. SAGEPublications. ISBN 978-1-4739-5323-9 (hardback), ISBN 978-1-4739-5324-6 (paperback), 978-1-5264-1729-9 (digital).

Reviewers would also like you to provide more details about your plans for:

Participant recruitment,Efforts to minimize potential harms to your participants,Planned interaction with your youth advisory board in terms of study design, data interpretation, and dissemination of your findings, andCompensation for the time spent on your project by your research participants and your youth advisory board.

I think your proposed study is important and will provide valuable insight for treatment and prevention efforts. I feel incorporating the feedback from our reviewers will further strengthen your study and the quality of the data you obtain. I look forward to reviewing your revision if you elect to revise and resubmit your protocol for further review.

Reviewers' comments:

Reviewer's Responses to Questions

**Comments to the Author**

1. Does the manuscript provide a valid rationale for the proposed study, with clearly identified and justified research questions?

Reviewer #1: Yes

Reviewer #2: Yes

Reviewer #3: Yes

2. Is the protocol technically sound and planned in a manner that will lead to a meaningful outcome and allow testing the stated hypotheses?

Reviewer #1: Yes

Reviewer #2: Partly

Reviewer #3: Yes

3. Is the methodology feasible and described in sufficient detail to allow the work to be replicable?

Reviewer #1: Yes

Reviewer #2: No

Reviewer #3: Yes

4. Have the authors described where all data underlying the findings will be made available when the study is complete?

Reviewer #1: No

Reviewer #2: No

Reviewer #3: No

5. Is the manuscript presented in an intelligible fashion and written in standard English?

Reviewer #1: Yes

Reviewer #2: Yes

Reviewer #3: Yes

6. Review Comments to the Author

You may also provide optional suggestions and comments to authors that they might find helpful in planning their study.

Reviewer #1: Thanks for taking on this research, it's important and sure to be impactful!

I'll attach my comments in a PDF.

Reviewer #2: See attachment for full review comments.

This is an exciting study that addresses an important gap in the literature and the authors intend to maximize TNG youth involvement. However, important details are missing processes for community engagement and how compensation is determined, especially due to the amount of labor they are requesting from TNG youth. The methods for qualitative analysis are also underdeveloped and inconsistent with how Braun & Clarke’s approach to thematic analysis has evolved. There is no mention of reflexivity as part of the analytic process, which is particularly concerning considering the nature of the study. Citations throughout are also relatively old and many could be updated. Finally, more attention could be paid to language throughout. I am curious if excluding intersex youth from TNG is on purpose or an oversight. If the study intends to include intersex youth please modify sociodemographic information collected.

Reviewer #3: The authors provide a well-planned study to address an area of need related to trans youth with DEBs. Overall, the protocol outlines a future study that could be very valuable and offer great progress in the field. There are a few areas that I believe could be strengthened and/or points that may need to be considered, as detailed below:

- I would encourage the authors to consider the age range for inclusion criteria for the study in Phases 2-4 and/or further explain the rationale for the chosen upper age limit. Because much of the introduction is specifically focused on TNG minors and many DEB scales begin at 18 for adult versions, it is unclear how the limit of 21 was chosen in how the authors define “youth”.

- Although Phase 2 is an important aspect of the study, it seems to be less connected to the other phases. Is there a reason the authors are interested in general barriers to TNG youth participation in research and not barriers to research for TNG youth with disordered eating? Might TNG youth experiencing DEBs have additional or specific barriers to participation that TNG youth without disordered eating may not have that would not be captured in Phase 2?

- The authors discuss some barriers to research participation for TNG youth, particularly parental consent and non-supportive caregivers. Given the high risk of safety concerns for TNG youth that may be outed by any potential breach of confidentiality, the authors are asked to address how they plan to safeguard participant identity and anonymity, particularly during qualitative interviews.

- Given community-based recommendations against collecting sex assigned at birth from trans participants unless absolutely necessary, the authors are asked to consider how they will collect demographic data and the extent to which sex assigned at birth versus gender is necessary for ensuring measurement invariance.

- The authors are encouraged to discuss any established plans for future changes to the study protocol after the YAB is established and how input from the YAB on study design will be solicited and incorporated prior to moving forward with Phases 2-4.

- Since the authors are planning to ask about medical affirmation during the sociodemographic survey, they are also encouraged to ask about participants’ desire to access such medical care. Not all trans and genderqueer individuals desire or seek out gender affirming medical care, so it may be important to understand medical affirmation in the context of each participant’s desire for such care.

7. PLOS authors have the option to publish the peer review history of their article (what does this mean?). If published, this will include your full peer review and any attached files.

Reviewer #1: **Yes: **Scout Silverstein

Reviewer #2: No

Reviewer #3: **Yes: **Bek Urban

---

## [Author Response · Author response to Decision Letter 0]

20 Sep 2024

ABSTRACT

1. Reviewer #1: “Transgender, non-binary, and gender-diverse (TNG) youth experience disordered eating behaviors (DEBs) for reasons unique to their gender identity” I recommend shifting this to have the onus be placed on the treatment of TNG youth, internalized gendered appearance ideals, internalized cisheteronormativity, etc. A TNG individual’s unique gender identity is in itself not inherently a risk factor. Additionally from Reviewer #2: Line 5- “for reasons unique to their gender identity”- This phrase is misleading, this insinuates that eating disorders are inherent to gender identity rather due to larger factors such as discrimination, marginalization, Euro-centric gendered beauty ideals, and lack of access to gender-affirming medical care. I recommend revising. 

a. We thank both reviewers for their thoughtful recommendations and we agree that risk factors for DEBs in TNG youth are not inherent to gender identity, or even gender dysphoria, but external factors placed on TNG youth. The sentence (lines 4-7) now reads, “Transgender, non-binary, and gender-diverse (TNG) youth experience disordered eating behaviors (DEBs) for reasons unique to their sociocultural positioning and the specific challenges they face, including societal beauty expectations of gender, transphobic discrimination, and lack of access to gender affirming medical care.”

2. Reviewer #2: Line 7- Recommend removing “vulnerable”—vulnerable to what? TNG youth are also inspiring, determined, and strong. This language is not helpful and does not add to this sentence. 

a. We appreciate the reviewer’s thoughtfulness around affirming and supportive language when discussing TNG youth. We have deleted this word from the sentence in line 9. 

3. Reviewer #2: Line 10- TNG populations are generally excited to participate in research with research institutions/individuals that they trust because they recognize TNG people have been excluded from research and much needs to be done to improve care for TNG people. Hesitations to engage in research are typically due to fear for safety/confidentiality which is not discussed here. It also may be helpful to specify ED research rather than speaking to research overall. See Burnette et al., 2022 re: demographics included in eating disorder treatment trials. 

a. We have added “DEBs research” to the end of this sentence (lines 9-12) to specify that these are gaps in this research area. 

b. Lines 10-13 provide a summary of gaps in research. We provide greater details on underrepresentation of TNG participants in DEBs research in the 3rd paragraph of the introduction section. 

INTRODUCTION

1. Reviewer #1: “TNG youth with eating disorder symptomatology are at the intersection of multiple, life-threatening risk factors.” This sentence is a bit circular - risk factors for what? Throughout the paper, there is little to no mention of DEBs resulting in physical health impairments. The closing sentence of this paragraph mentions multiple health disparities. Perhaps this is a good place to introduce content about that. Additionally, Reviewer #2: Line 27-28- you say multiple “life-threatening risk factors”, but only list suicidality? 

a. We have deleted this sentence completely given multiple reviewers noted that it was confusing. We also added examples of physical and emotional impairments. This section (lines 28-35) now reads, “Based upon data from the general population, DEBs are associated with significant impairment (i.e., changes in menstruation and fertility, substance use, depressive symptoms) and often precede the onset of eating disorders. Furthermore, 30-51% of TNG youth report suicidal ideation, and TNG young adults with eating disorders attempt suicide at higher rates than their TNG peers without eating disorders. Despite these health disparities among TNG youth, DEBs and eating disorder research has historically focused on white, cisgender young adult women.”

2. Reviewer #1: “No current measures assess the attitudes and cognitions underlying an individual TNG youth’s DEBs. This creates a major barrier to culturally specific care. A tailored instrument that assesses the specific behaviors, attitudes, and cognitions contributing to TNG youths’ DEBs is needed to advance research on this understudied topic and improve clinical care.” I would encourage mention of underdiagnosis specifically, as it is most directly and immediately related to creating a screening and/or assessment tool.

a. We agree with the reviewer and these sentences (lines 41-43) now read, “No current measures assess the attitudes and cognitions underlying an individual TNG youth’s DEBs. This creates a major barrier to gender-affirming care and accurate diagnosis of EDs and DEBs in trans youth.”

3. Reviewer #1: “Although the field of TNG DEBs research is quickly growing, gaps in knowledge remain, due, in part, to a lack of TNG input on research protocols focused on this population, and underrepresentation of TNG participation in research.” This is an excellent point. I would encourage expanding on this briefly to indicate how the lack of TNG input frequently translates into flawed methodologies. It feels loosely inferred in lines 62-71, but stating this more explicitly may be more impactful.

a. Thank you for the opportunity to provide more details on this topic. We have added the sentences (lines 64-67), “Race/ethnicity, access to resources, and gender identity have an impact on TNG youth experiences. As a result, if researchers are unable to recruit samples that represent TNG youth diversity, validity can be compromised” 

4. Reviewer #1: The paragraph discussing underrepresentation of TNG participants in research and barriers to participation (lines 43-61) may be strengthened by including a mention of state surveillance and safety risks of disclosure. This is very briefly mentioned in lines 253-255, but merits being introduced earlier. Also consider highlighting rural and/or southern TNG youth as potentially being underrepresented in research – confirm with a literature search.

a. We have added lines 52-53, “Moreover, the increase in anti-transgender legislation over the past 5 years may affect TNG youth research participation.”

b. In line 56, we now specify oversampling of TNG youth with access to gender-affirming care in urban areas. 

5. Reviewer #2: Line 20-22- It is important to mention that vulnerability to body dissatisfaction among youth may also be exacerbated due to lack of access to gender-affirming medical interventions. Currently this is only mentioned in the discussion. The way this is written now it makes it seem like DEBs among TNG youth is an inherent part of pubertal development. Also, there is no mention of economic disparities (due to structural discrimination/marginalization) among trans populations and how that impacts DEBs. 

a. We agree that access to gender-affirming care may affect body dissatisfaction associated with pubertal changes. This sentence (lines 22-26) has been modified to, “TNG youth also have increased vulnerability to body dissatisfaction due to pubertal changes and development of secondary sexual characteristics that might be misaligned with their gender identity, which may be exacerbated by a youth’s inability to access gender-affirming medical care (i.e., puberty blockers, gender-affirming hormones).”

b. We only found one article describing socioeconomic status and DEBs/eating disorders and it is a qualitative study (https://pubmed.ncbi.nlm.nih.gov/39102353/). We have added measures of socioeconomic status to Phase 4 surveys to better understand this relationship. 

6. Reviewer #2: Line 31- remove “significantly increased”—this is redundant based on prior sentence. 

a. We have deleted “significantly increased” from this sentence (lines 33-35).

7. Reviewer #2: Line 52- accurate is said twice in this sentence, I recommend removing the first “accurate”

a. Thank you for the opportunity to correct this error. We have deleted the first “accurate” in this sentence (lines 58-64).

8. Reviewer #2: Line 55-56- this is old data (particularly Arcelus), I’d recommend updating citation or mentioning here that recent estimates are not available. It seems odd to me to focus so much on percentages of trans masc vs trans femme vs nonbinary/gender diverse, especially as these categories are not exclusive, for example many people are trans masc or trans femme and nonbinary. It is important to focus on transfemme recruitment due to underrepresentation but I would not make it as much of a central point, especially because the data cited is not recent and there have been evolutions in how we collect gender identity information. 

a. Thank you for the opportunity to provide more up to date estimates of TNG people by gender identity. We have provided estimates from a 2022 report (Reference #25).

b. We agree that measuring gender identity is a complex process and we describe issues with estimates in the sentence (lines 57-64), “Approximately 38.5% of TNG adults are transfeminine, 35.9% transmasculine, and 25.6% gender non-conforming[24]. Although estimates of TNG population are not fully accurate due to the broad range of terms surveys use to describe gender identity, lack of TNG data collection in population-based surveys, and difficulty creating gender identity categories (as individuals can identify with multiple gender identities), this discrepancy between the proportion of transfeminine people and their underrepresentation in research is a critical gap in knowledge because transfeminine people, particularly transfeminine people of color, experience unique physical and mental health challenges.” We acknowledge that research has evolved with respect to measurement of gender identity in research, and we are only able to report proportions based on the individual study’s methods even if outdated at present. We have chosen to keep this information in the manuscript because we feel it is important to describe the difference in representation of gender identities in research samples versus population estimates of gender identity. 

9. Reviewer #2: Line 59- these citations are also old (2006-2009), there have been many studies about health disparities among trans populations published in the past five years. One example is: Hughes et al., 2022- which looks at mortality rates. 

a. Thank you for directing our attention to more recent literature. We now include more recent citations, such as the Hughes 2022 article recommended by the reviewer (References #26-29).

10. Reviewer #2: Line 69- “culturally specific” could be deleted here. Overall “culturally specific” is used frequently. However, it is not defined what this means. There are many cultures within TNG communities (stated in the following sentence), so this use throughout without defining what is meant by the term “cultural” is strange and communicates assumptions about monolithic trans experience. 

a. Thank you for this thoughtful recommendation. We have deleted “culturally specific” and “culturally sensitive” throughout the manuscript.

11. Reviewer #2: Line 70-71- Why is trans femme excluded here? When the authors spent a while talking about trans femme exclusion earlier in the introduction? 

a. We have added transfeminine individuals to this sentence (lines 76-79). 

MATERIALS AND METHODS

Methods

1. Reviewer #1: Compensation for YAB is mentioned in line 268. However, it is mentioned that compensation will occur quarterly, while elsewhere in the paper, it reads that YAB will meet twice monthly. Please provide a rationale for this and detail your plan for compensating research participants outside of the YAB.

a. Thank you for the opportunity to correct this error and add these details. We have added a column in Table #2 that provides the compensation amount for each research activity.

2. Reviewer #1: “In addition to confirming scale structure, measurement invariance in relation to sex assigned at birth, gender identity, and race/ethnicity will be evaluated. Based on extant literature, we will also explore associations between DEBs and TNG-specific risk and protective factors (i.e., mental health diagnoses, gender minority stress, peer/family support, access to affirming medical care).” Have you considered incorporating socioeconomic status? There is some great research highlighting the link between food insecurity and DEBs, and showing increased food insecurity in TNG populations.

a. We thank the reviewer for this important recommendation. We have added measures of socioeconomic status (education, income, food insecurity) to the data that will be collected in Phase 4.

3. Reviewer #2: Line 78- isn’t it disordered eating behaviors and related attitudes and cognitions? Something just assessing DEB prevalence would be a different measure and this study’s interest in attitudes and cognitions underlying DEBs is unique. I would recommend being consistent throughout. 

a. We have edited the sentence (lines 84-85) to read, “This research protocol will use a community-engaged approach to create a TNG youth-centered measure of disordered eating behaviors, attitudes, and cognitions.”

4. Reviewer #2: Line 79-80- how will the YAB be recruited? I see that later they will be compensated quarterly, but how is compensation determined? This is a lot of labor for a YAB. 

a. Recruitment details for the YAB can be found in lines 159-165. 

b. Thank you for the opportunity to provide clarifying information on YAB compensation. We have added lines 87-94 to describe the responsibilities and time commitment for active participation as a YAB member. 

5. Reviewer #2: Line 86- Thematic saturation may not be a feasible goal (or a helpful metric) for this type of thematic analysis. See: Braun V, Clarke V. To saturate or not to saturate? Questioning data saturation as a useful concept for thematic analysis and sample-size rationales. Qualitative Research in Sport, Exercise and Health. 2021;13(2):201-216. doi:10.1080/2159676X.2019.1704846

a. We thank the reviewer for providing their expertise on thematic analysis. We have read the recommended reading and put more thought into our sample size for Phase 2. This section (lines 108-112) now reads, “We will oversample underrepresented TNG identities (i.e., TNG youth of color, transfeminine and non-binary youth). We have chosen a sample size of 20 participants based on the focus of the research question, methods of data collection (see below), sample diversity, and depth of data likely generated from each participant.”

6. Reviewer #2: Line 90-94- Have the authors considered any particular facilitation strategy for these meetings? Co-design strategies or concept mapping are some examples of facilitation strategies designed to maximize meaningful engagement with YABs. Have facilitators received any training on YAB facilitation? 

a. We thank the reviewer for the opportunity to expand on our facilitation approach to YAB participation/meetings. We have added lines 94-104.

7. Reviewer #2: Line 97- throughout “patterns of cognitive processing problems” is used. This reads as pathologizing rather than indicating issues with the measure itself. You could just say patterns of cognitive processing rather than naming it as problems. 

a. Thank you to the reviewer for their dedication to affirming language. We have deleted “problems” from line 126.

8. Reviewer #2: Line 115- Recruitment- It felt odd to put recruitment information after the stages, I would think it would make more sense to put recruitment information upfront or integrated throughout the Phases

a. In the current format, we provide a summary of each phase at the beginning of the Materials and methods section and then have the subsections: recruitment, inclusion/exclusion, data collection, and analyses; starting with recruitment. Because there are multiple phases and similar processes across different phases in all subsections (i.e., similar recruitment practices for Phases 2-4, similar qualitative data collection and analyses in Phases 2 and 3), we organized the Materials and methods section by these subsections instead of Phases to eliminate redundancy. 

9. Reviewer #2: Line 141- How will history of DEBs be assessed for those participating in the advisory board? 

a. Thank you for the opportunity to include this 

---

## [Decision Letter · Decision Letter 1]

26 Oct 2024

PONE-D-24-25313R1Using youth-engaged research methods to develop a measure of disordered eating in transgender, non-binary, and gender-diverse youth: research protocolPLOS ONE

Dear Dr. Pham,

Thank you for submitting your revised protocol to PLOS ONE. I think it is much improved and I am eager to see the results of your study. However, there are a four suggestions raised by the reviewers that I would like you to consider.

**Please address the following structural concerns with your protocol:**

1. Compensation of research participants for their time:

“provide rationale for compensating members of the YAB in a lump-sum every 6 months as opposed to after every meeting, especially given the prevalence of financial disparities for trans youth and the length of time between meetings and payment. . . . consider implications for retention of YAB members and how financial barriers may result in a lack of diversity among the YAB if compensation for time provided is only awarded twice per year.”

2. Plan for Human Subjects Protections:

“provide any additional information related to planned recording of the interviews, methods and timeline for transcription of recordings, and how/when interview transcripts will be de-identified and unlinked from recordings”

**Please consider how your research participants, potential funders, reviewers, readers, will perceive the following decisions:**

3. Use of the more common term "transphobia" versus the less common but more accurate cis-sexism:

“I would encourage the authors to move away from the use of "transphobic discrimination" to "cissexism", as the latter term explicitly names the system of oppression driving discrimination against trans individuals and not a phobia of trans people.”

4: Collection of sex assigned at birth in a study of transgender and gender-diverse individuals who do not identify with their sex assigned at birth:

"I would encourage the authors to continue considering the need for collection of sex in Phase 4, in combination with Reviewer 1's suggestion to collect data regarding intersex status. Although measurement invariance is important, it remains unclear to me if asking participants about their sex assigned at birth is meaningful enough to offset potential impacts and not honor community-based recommendations. Will testing invariance by sex be limited to a male/female binary, and if so, how do the authors plan to include intersex individuals? Finally, do the authors have other theoretical rationale for including sex assigned at birth aside from capacity to menstruate? Due to variation in onset of puberty, use of birth control, and other factors, not all individuals who are assigned female at birth can or do menstruate. If the concern about measurement variance is only tied to menstruation, I would encourage authors to consider how to collect information about menstruation specifically."

You do not need to change your protocol based on suggestions 3 and 4, but I would like you to give thought to the feedback from the reviewers on these decisions. The full text of the reviewer's feedback is listed below. Please include the following items when submitting your revised protocol:A rebuttal letter that responds to each point raised by the academic editor. You should upload this letter as a separate file labeled 'Response to Reviewers'.A marked-up copy of your manuscript that highlights changes made to the original version. You should upload this as a separate file labeled 'Revised Manuscript with Track Changes'.An unmarked version of your revised paper without tracked changes. You should upload this as a separate file labeled 'Manuscript'.

Kind regards,

Christina M. Roberts, M.D., M.P.H.

Academic Editor

PLOS ONE

Reviewers' comments:

Reviewer's Responses to Questions

**Comments to the Author**

1. Does the manuscript provide a valid rationale for the proposed study, with clearly identified and justified research questions?

Reviewer #2: Yes

Reviewer #3: Yes

2. Is the protocol technically sound and planned in a manner that will lead to a meaningful outcome and allow testing the stated hypotheses?

Reviewer #2: Yes

Reviewer #3: Yes

3. Is the methodology feasible and described in sufficient detail to allow the work to be replicable?

Reviewer #2: Yes

Reviewer #3: Yes

4. Have the authors described where all data underlying the findings will be made available when the study is complete?

Reviewer #2: Yes

Reviewer #3: Yes

5. Is the manuscript presented in an intelligible fashion and written in standard English?

Reviewer #2: Yes

Reviewer #3: Yes

6. Review Comments to the Author

You may also provide optional suggestions and comments to authors that they might find helpful in planning their study.

Reviewer #2: The authors have significantly improved the manuscript by providing further detail on methods, further engaging with the literature, and providing further details.

There are several areas where the language is not as concise as it could be (e.g. using "as a result of" instead of "due to", "not recreate" instead of "avoid"), and sentences are quite long/meandering e.g., lines 69-75 (could be changed to something like "although prevalence rates of gender diverse identities vary significantly across surveys, trans femme people are underrepresented in eating disorder research (cite). It is important to address this gap because..."). I'm also not sure about the language "biased towards" in line 173. I recognize that some of this is a style preference, but I do think that further editing could improve readability throughout.

Reviewer #3: I appreciate the changes made by the authors and their responsiveness to reviews. I have a few other points of feedback for the authors:

Abstract

1. In response to other reviewers, the authors made edits in lines 4-7. The edits more clearly frame factors related DEB among TNG youth. I would encourage the authors to move away from the use of "transphobic discrimination" to "cissexism", as the latter term explicitly names the system of oppression driving discrimination against trans individuals and not a phobia of trans people.

Materials and Methods

1. The authors added information about payment for each phase of the study. Phase 1 notes that the YAB will be paid $250 for every 6 months of participation. The authors are asked to provide rationale for compensating members of the YAB in a lump-sum every 6 months as opposed to after every meeting, especially given the prevalence of financial disparities for trans youth and the length of time between meetings and payment. The authors are also asked to consider implications for retention of YAB members and how financial barriers may result in a lack of diversity among the YAB if compensation for time provided is only awarded twice per year.

2. I thank the authors for adding information about protecting interviewees. I would encourage the authors to provide any additional information related to planned recording of the interviews, methods and timeline for transcription of recordings, and how/when interview transcripts will be de-identified and unlinked from recordings, as recordings carry the highest level of risk for participants.

Inclusion/Exclusion criteria:

1. Thank you for this update. The authors are still asked to respond to how studies reviewed in the introduction that focus on minors apply to TNG individuals who are 18-21 and any additional considerations for this age group that is to be included in the study.

Data collection

1. I would encourage the authors to continue considering the need for collection of sex in Phase 4, in combination with Reviewer 1's suggestion to collect data regarding intersex status. Although measurement invariance is important, it remains unclear to me if asking participants about their sex assigned at birth is meaningful enough to offset potential impacts and not honor community-based recommendations. Will testing invariance by sex be limited to a male/female binary, and if so, how do the authors plan to include intersex individuals? Finally, do the authors have other theoretical rationale for including sex assigned at birth aside from capacity to menstruate? Due to variation in onset of puberty, use of birth control, and other factors, not all individuals who are assigned female at birth can or do menstruate. If the concern about measurement variance is only tied to menstruation, I would encourage authors to consider how to collect information about menstruation specifically.

7. PLOS authors have the option to publish the peer review history of their article (what does this mean?). If published, this will include your full peer review and any attached files.

Reviewer #2: No

Reviewer #3: **Yes: **Bek Urban

---

## [Author Response · Author response to Decision Letter 1]

30 Oct 2024

OVERALL

1. Reviewer #2: There are several areas where the language is not as concise as it could be (e.g. using "as a result of" instead of "due to", "not recreate" instead of "avoid"), and sentences are quite long/meandering e.g., lines 69-75 (could be changed to something like "although prevalence rates of gender diverse identities vary significantly across surveys, trans femme people are underrepresented in eating disorder research (cite). It is important to address this gap because..."). I'm also not sure about the language "biased towards" in line 173. I recognize that some of this is a style preference, but I do think that further editing could improve readability throughout.

a. Thank you for your feedback. We have made the following edits:

i. Replaced “as a result of” with “due to” in line 21.

ii. Replaced “not recreate” with “avoid” in line 169.

iii. Replaced “has been biased towards” with “has primarily sampled” in line 163.

iv. Regarding the original sentence, “Although estimates of TNG population are not fully accurate due to the broad range of terms surveys use to describe gender identity, lack of TNG data collection in population-based surveys, and difficulty creating gender identity categories (as individuals can identify with multiple gender identities), this discrepancy between the proportion of transfeminine people and their underrepresentation in research is a critical gap in knowledge because transfeminine people, particularly transfeminine people of color, experience unique physical and mental health challenges[26-29].”, we did not simplify per the recommendation of the reviewer because we feel it is important that the reader learn about the reasons for inaccurate estimates. We agree with the reader that the original sentence was difficult to read because of its length and we have separated it into 2 separate sentences, “Estimates of TNG population are not fully accurate due to the broad range of terms surveys use to describe gender identity, lack of TNG data collection in population-based surveys, and difficulty creating gender identity categories (as individuals can identify with multiple gender identities). Despite this, the discrepancy between the proportion of transfeminine people and their underrepresentation in research is a critical gap in knowledge because transfeminine people, particularly transfeminine people of color, experience unique physical and mental health challenges[26-29].” (lines 58-64)

ABSTRACT

1. Reviewer #3: In response to other reviewers, the authors made edits in lines 4-7. The edits more clearly frame factors related DEB among TNG youth. I would encourage the authors to move away from the use of "transphobic discrimination" to "cissexism", as the latter term explicitly names the system of oppression driving discrimination against trans individuals and not a phobia of trans people.

a. Thank you for suggesting a more up to date term. We have replaced “transphobic discrimination” with “cissexism” in line 7.

MATERIALS AND METHODS

Methods

1. Reviewer #3: The authors added information about payment for each phase of the study. Phase 1 notes that the YAB will be paid $250 for every 6 months of participation. The authors are asked to provide rationale for compensating members of the YAB in a lump-sum every 6 months as opposed to after every meeting, especially given the prevalence of financial disparities for trans youth and the length of time between meetings and payment. The authors are also asked to consider implications for retention of YAB members and how financial barriers may result in a lack of diversity among the YAB if compensation for time provided is only awarded twice per year.

a. Thank you for this important comment. While our first resubmission was under review, the research team met with community partners and other researchers with TNG YAB experience to discuss and further refine our YAB compensation plan. After receiving feedback, we decided to provide more frequent compensation. We will be providing $125 for every 3 months of participation instead of $250 for every 6 months of participation. We are not providing compensation after each meeting because members may not be able to attend every meeting. Additionally, after receiving feedback, we will be providing $20 for each transcript that YAB members code in Phases 2 and 3. These changes are described in lines 92-95.

2. Reviewer #3: I thank the authors for adding information about protecting interviewees. I would encourage the authors to provide any additional information related to planned recording of the interviews, methods and timeline for transcription of recordings, and how/when interview transcripts will be de-identified and unlinked from recordings, as recordings carry the highest level of risk for participants.

a. Thank you for the opportunity to add additional details to our data collection for qualitative interviews in Phases 2 and 3. We have added lines 217-221 and 224-227.

Inclusion and exclusion criteria

1. Reviewer 3: Thank you for this update. The authors are still asked to respond to how studies reviewed in the introduction that focus on minors apply to TNG individuals who are 18-21 and any additional considerations for this age group that is to be included in the study.

a. Most of the studies cited in the introduction describe TNG youth, including participants up to 25 years of age. Other than consent processes, we do not have different protocols for participants 18 and older. For each phase, we are measuring age and thus, our findings may result in additional considerations for different age groups within TNG youth. 

Data collection

1. Reviewer #3: I would encourage the authors to continue considering the need for collection of sex in Phase 4, in combination with Reviewer 1's suggestion to collect data regarding intersex status. Although measurement invariance is important, it remains unclear to me if asking participants about their sex assigned at birth is meaningful enough to offset potential impacts and not honor community-based recommendations. Will testing invariance by sex be limited to a male/female binary, and if so, how do the authors plan to include intersex individuals? Finally, do the authors have other theoretical rationale for including sex assigned at birth aside from capacity to menstruate? Due to variation in onset of puberty, use of birth control, and other factors, not all individuals who are assigned female at birth can or do menstruate. If the concern about measurement variance is only tied to menstruation, I would encourage authors to consider how to collect information about menstruation specifically.

a. We thank the reviewer for their continued advocacy on affirming research procedures. We will be working closely with our community consultant, YAB, and community stakeholders for each Phase. Prior to Phase 4 data collection, we will assess the need to collect sex assigned at birth. We will follow the reviewer’s recommendations of considering the rationale, impacts, and inclusion of intersex people when discussing sex assigned at birth data collection and analysis with community members. We have added lines 257-258 to reflect that the list of measures will be finalized after receiving feedback from the community.

---

## [Editor Report · Decision Letter 2]

4 Nov 2024

Using youth-engaged research methods to develop a measure of disordered eating in transgender, non-binary, and gender-diverse youth: research protocol

PONE-D-24-25313R2

Dear Dr. Pham,

We’re pleased to inform you that your manuscript has been judged scientifically suitable for publication and will be formally accepted for publication once it meets all outstanding technical requirements.

Kind regards,

Christina M. Roberts, M.D., M.P.H.

Academic Editor

PLOS ONE
---

## [Editor Report · Acceptance letter]

8 Nov 2024

PONE-D-24-25313R2 

PLOS ONE

Dear Dr. Pham, 

I'm pleased to inform you that your manuscript has been deemed suitable for publication in PLOS ONE. Congratulations! Your manuscript is now being handed over to our production team.

Kind regards, 

on behalf of

Dr. Christina M. Roberts 

Academic Editor

PLOS ONE